# Does Quantum Mechanics Require “Conspiracy”?

**DOI:** 10.3390/e26050411

**Published:** 2024-05-09

**Authors:** Ovidiu Cristinel Stoica

**Affiliations:** Department of Theoretical Physics, NIPNE—HH, 077125 Bucharest, Romania; cristi.stoica@theory.nipne.ro or holotronix@gmail.com

**Keywords:** Born rule, fine-tuning, statistical independence, records, memories, arrow of time, Past Hypothesis, decoherence, Bell’s theorem, interpretations of quantum mechanics, superdeterminism

## Abstract

Quantum states containing records of incompatible outcomes of quantum measurements are valid states in the tensor-product Hilbert space. Since they contain false records, they conflict with the Born rule and with our observations. I show that excluding them requires a fine-tuning to an extremely restricted subspace of the Hilbert space that seems “conspiratorial”, in the sense that (1) it seems to depend on future events that involve records (including measurement settings) and on the dynamical law (normally thought to be independent of the initial conditions), and (2) it violates Statistical Independence, even when it is valid in the context of Bell’s theorem. To solve the puzzle, I build a model in which, by changing the dynamical law, the same initial conditions can lead to different histories in which the validity of records is relative to the new dynamical law. This relative validity of the records may restore causality, but the initial conditions still must depend, at least partially, on the dynamical law. While violations of Statistical Independence are often seen as non-scientific, they turn out to be needed to ensure the validity of records and our own memories and, by this, of science itself. A Past Hypothesis is needed to ensure the existence of records and turns out to require violations of Statistical Independence. It is not excluded that its explanation, still unknown, ensures such violations in the way needed by local interpretations of quantum mechanics. I suggest that an as-yet unknown law or superselection rule may restrict the full tensor-product Hilbert space to the very special subspace required by the validity of records and the Past Hypothesis.

## 1. Introduction

Quantum mechanics, like other theories, is formulated from a God’s-eye perspective. But, as parts of the world we observe, we are limited to a worm’s-eye perspective. If, in the present time, we are part of a random state of the universe, this would most likely contain incompatible records from which we would never be able to guess the laws of quantum mechanics, particularly the Born rule.

An example of such a state is one containing *n* records of repeated spin measurement of the same silver atom so that the records of the *n* outcomes are random values $\pm \frac{1}{2}$, and not the same value repeated *n* times. This state is valid in the tensor-product Hilbert space. But the records it contains could not come from actual repeated quantum measurements. We practically never observe such states.

The simple fact that we exist and could discover quantum mechanics indicates that the physical law is user-friendly enough to allow our memories to form and be reliable, to reflect the evolution of our universe so that we can guess its laws, including the Born rule. We are led to a *“the universe does not mislead us”* metaprinciple:

**Metaprinciple** **NMU**(Non-Misleading Universe). *The records of the experimental results and the memories of the observers reflect the actual history of the universe.*

Without this, science and even life would be impossible. But Metaprinciple NMU, as we shall see, requires severe restrictions of the possible states. We will explore the relationship between this fine-tuning and several common-sense beliefs. The first belief is:

**Belief** **1**(Universality)**.**
*Quantum mechanics, including the Born rule and the results of quantum experiments, respect Metaprinciple NMU for most initial conditions.*

Another belief is that of *Statistical Independence* (SI). We assume that there are enough degrees of freedom so that any two systems separated in space can be put in independent and statistically uncorrelated states.

**Definition** **1.**
*Two events A and B are statistically independent if Pr{AB}=Pr{A}Pr{B} ([1] p. 10). In particular, if each of two statistically independent events A and B are possible (Pr{A}>0 and Pr{B}>0), they are possible together (Pr{AB}>0). Therefore, if SI is true for the events that two subsystems are in particular states, the following should be true as well:*


**Belief** **2**(Subsystem Independence)**.**
*Let A and B be any two subsystems with no common parts. If A can possibly be in the state α and B can possibly be in the state β, the combined system can possibly be in the state α⊗β.*

Belief 2 is the core reason we take as the Hilbert space of a composite system the tensor product of the Hilbert spaces of each of the systems. I will show that this, and consequently SI, is contradicted, although *Bell’s weaker assumption of SI is not contradicted* (see Answer 5). Bell’s assumption is about the independence of the state of the observed system (including possible hidden variables) of the choice of what observable will be measured [2]. It is sometimes called *settings independence, measurement independence,* or even the *free-will assumption* [3].

The following belief is already known to be violated for “Boltzmann brains”, usually attributed to fluctuations:

**Belief** **3**(For-Granted Memory)**.**
*In the tensor-product Hilbert space formulation of quantum mechanics, past events always leave reliable records in the present state.*

It makes sense to think that no “Laplace demon” knowing the dynamical law and the future histories is needed to determine what initial conditions ensure Metaprinciple NMU. This can be stated as the following beliefs:

**Belief** **4**(No Input From Dynamical Law)**.**
*Initial conditions are independent of the dynamical law of the system.*

**Belief** **5**(No Input From Future)**.**
*Initial conditions are independent of future events in history, in particular of those involving records.*

In Section 2, I show that, in quantum mechanics, Metaprinciple NMU contradicts Beliefs 1, 2, and 3, and seems to challenge Beliefs 4 and 5.

In Section 3, I gradually build a model that shows that Belief 4 is partially not contradicted, and Belief 5 is not contradicted at all, at least for interpretations that do not violate Bell’s restricted Statistical Independence. It turns out that for this to work, quantum mechanics needs to include a certain sophistication, and the Past Hypothesis, according to which the initial state of the universe had a very low entropy, is required in a way that involves the dynamical law.

In Section 4, I discuss the implications for the Past Hypothesis.

In Section 5, I discuss the violation of Statistical Independence resulting from the Theorem and its relationship with the particular case used by Bell in his theorem.

In Section 6, I discuss the possibility that a new law may solve the puzzle.

## 2. The Puzzle

In this Section, I state the puzzle in the form of a Theorem. In Section 3, I will try to build a model to resolve the puzzle and succeed only partially.

**Theorem** **1.**
*Metaprinciple NMU for quantum mechanics requires the initial states to belong to an extremely restricted subspace of the Hilbert space in a way that contradicts Beliefs 1, 2, and 3, and seems to challenge Beliefs 4 and 5 (which I will try to restore in Section 3).*


**Proof.** Consider a closed quantum system which includes observed systems, measuring devices, and observers. This may be the entire universe. Its states are represented by unit vectors in a separable Hilbert space H, and evolve governed by the Schrödinger equation with the Hamiltonian $\hat{H}$. In terms of the unitary evolution operator ${\hat{U}}_{t,t_{0}}:=e^{-\frac{i}{\hbar }\left(t-t_{0}\right)\hat{H}}$ between the times $t_{0}$ and *t*, the evolution of an initial state vector $\Psi (t_{0})\in H$ at $t_{0}$ is
(1)$$\Psi \left(t\right)={\hat{U}}_{t,t_{0}}\Psi \left(t_{0}\right).$$Suppose for simplicity that our system contains a system to be observed *S*, with Hilbert space $H_{S}$ of finite dimension $\dim H_{S}=n<\infty$, and a measuring device whose pointer is represented in the Hilbert space $H_{M_{A}}$ of dimension n+1. Then $H=H_{S}⊗H_{M_{A}}⊗H_{E}$, where $H_{E}$ represents everything else, including the other parts of the measuring device. Let $\hat{A}$ be a Hermitian operator on $H_{S}$ representing the observable of interest, with eigenbasis $\left(\psi_{1}^{A},\ldots ,\psi_{n}^{A}\right)$. Let ${\hat{Z}}^{A}$ be the pointer observable, with eigenbasis $\left(\zeta_{0}^{A},\zeta_{1}^{A},\ldots ,\zeta_{n}^{A}\right)$, where $\zeta_{0}^{A}$ represents the “ready” state of the pointer. We assume that the observable and the pointer have nondegenerate spectra, and the measurement is ideal. We work in the interaction picture, which allows us to treat the observed degrees of freedom as stationary and the pointer states as stationary before and after the measurement. Let the measurement of $\hat{A}$ take place between $t_{0}$ and $t_{1}>t_{0}$, leading to the superposition
(2)$$\Psi \left(t_{1}\right)={\hat{U}}_{t_{1},t_{0}}\psi ⊗\zeta_{0}^{A}⊗\ldots =\sum_{j}\langle \psi_{j}^{A}|\psi \rangle \psi_{j}^{A}⊗\zeta_{j}^{A}⊗\ldots$$To resolve the superposition from Equation (Equation 2) into definite outcomes, one usually invokes projection, objective collapse, decoherence into branches, additional hidden variables, etc.The results from this article apply to all these options. The *Born rule* states that the probability that at $t_{1}$ the pointer is in the state $\zeta_{j}^{A}$ is $|\langle \psi_{j}^{A}|\psi \rangle {|}^{2}$.Consider a second measurement of an observable $\hat{B}$ of the system *S*, with eigenbasis $\left(\psi_{1}^{B},\ldots ,\psi_{n}^{B}\right)$. Let the pointer observable of the second apparatus be ${\hat{Z}}^{B}$, with eigenbasis $\left(\zeta_{0}^{B},\zeta_{1}^{B},\ldots ,\zeta_{n}^{B}\right)$, where $\zeta_{0}^{B}$ is the “ready” state. The total Hilbert space is now $H=H_{S}⊗H_{M_{A}}⊗H_{M_{B}}⊗H_{E}$.The measurement of $\hat{B}$ takes place after the first measurement, ending before $t_{2}>t_{1}$. It leads to
(3)$$\begin{array}{cc} \Psi (t_{2})\; & ={\hat{U}}_{t_{2},t_{1}}{\hat{U}}_{t_{1},t_{0}}\psi ⊗\zeta_{0}^{A}⊗\zeta_{0}^{B}⊗\ldots \\ \; & ={\hat{U}}_{t_{2},t_{1}}\sum_{j}\langle \psi_{j}^{A}|\psi \rangle \psi_{j}^{A}⊗\zeta_{j}^{A}⊗\zeta_{0}^{B}⊗\ldots \\ \; & =\sum_{j,k}\langle \psi_{j}^{A}|\psi \rangle \langle \psi_{k}^{B}|\psi_{j}^{A}\rangle \psi_{k}^{B}⊗\zeta_{j}^{A}⊗\zeta_{k}^{B}⊗\ldots \end{array}$$The probability that at $t_{2}$ the first pointer state is $\zeta_{j}^{A}$ and the second pointer state is $\zeta_{k}^{B}$ is $|\langle \psi_{j}^{A}|\psi \rangle \langle \psi_{k}^{B}|\psi_{j}^{A}\rangle {|}^{2}$. This vanishes if $\hat{A}=\hat{B}$ and j≠k, and we obtain**Observation** **1.**
*The Born rule forbids orthogonal results for repeated measurements, e.g., if $\hat{A}=\hat{B}$, the states $\psi_{k}^{B}⊗\zeta_{j}^{A}⊗\zeta_{k}^{B}⊗\ldots$ with j≠k are practically forbidden at $t_{2}$, in the sense that to ensure the Born rule, such states must be extremely rare.*
**Observation** **2.**
*However, a priori, all unit vectors in H are possible initial conditions at the initial time $t_{i}<t_{0}$ of the universe, including, for any j and k, the vectors*

(4)
$$\Psi_{j,k}\left(t_{i}\right):={\hat{U}}_{t_{2},t_{i}}^{†}\psi_{k}^{B}⊗\zeta_{j}^{A}⊗\zeta_{k}^{B}⊗\ldots$$


*The uniform probability distribution on the projective Hilbert space gives equal probabilities to all possible states $\Psi_{j,k}\left(t_{i}\right)$ at any time $t_{i}$, but the Born rule gives a totally different probability. If $\hat{A}=\hat{B}$ (or in general $\left[\hat{A},\hat{B}\right]=\hat{0}$), the probability vanishes for j≠k.*
In practice, quantum measurements are not exactly sharp, so states as in Observation 1 are not exactly forbidden, but in the case of repeated measurements with $\hat{A}=\hat{B}$ the probability distribution is concentrated on the states $\Psi_{j,j}\left(t_{i}\right)$ and is negligible for the other states. Because they contain misleading records, we will call them “misleading states”. Misleading states are possible, albeit extremely improbable.In general, including if $\left[\hat{A},\hat{B}\right]\neq \hat{0}$, the Born rule gives, for the state $\Psi_{j,k}\left(t_{i}\right)$, the probability $|\langle \psi_{j}^{A}|\psi \rangle \langle \psi_{k}^{B}|\psi_{j}^{A}\rangle {|}^{2}$, which again is very different from the uniform probability distribution on the projective Hilbert space. The difference is significant enough so that we could infer from experiments the Born rule and not any other probabilistic rule. In this case, the states $\Psi_{j,k}\left(t_{i}\right)$ are allowed even if j≠k, as long as the probability does not vanish, but the initial conditions are constrained statistically. One may hope that an initially uniform probability distribution evolves, converging towards the Born rule, perhaps by successive projections or decoherence. But if this were true, it should have been true for the limiting case $\hat{A}=\hat{B}$ as well. Therefore, the Born rule must come from a different initial probability distribution than the uniform one.**Observation** **3.**
*The problem is not whether a state containing measuring devices in the “ready” state and observed systems can evolve, by performing measurements, into misleading states as in Observation 1. This is not the case (see Answer 1). But such states are represented by valid vectors in the total tensor-product Hilbert space, and they can be reached by unitary evolution (even including projections) from states like (Equation 4). This evolution into misleading states avoids the natural course of events, but the resulting state contains records of false accounts of history. They can even contain multiple pieces of camera footage from different angles, and human observers with false memories to attest to the false history. Nothing seems to prevent this since we can only access the present, not the past. So, the problem is that there is nothing in the formulation of quantum mechanics that makes misleading states extremely improbable.*
**Refutation** **1**(of Belief 1). *This part of the analysis depends on the approach to resolve the superposition into definite outcomes. We consider first unitary approaches based on decoherence (like the consistent histories approach [4] and the many-worlds interpretation (MWI) [5]). From Observation 1, initial states $\Psi_{j,k}\left(t_{i}\right)$ with $\hat{A}=\hat{B}$ and j≠k are misleading states, because they evolve into misleading states at $t_{2}$. Such states would contradict Metaprinciple NMU. Moreover, all initial states that are not orthogonal on all misleading states of the form $\Psi_{j,k}\left(t_{i}\right)$ should also be extremely unlikely because the nonvanishing component $\Psi_{j,k}\left(t_{i}\right)$ leads to a misleading branch with nonzero amplitude. Therefore, the states that are not initially misleading have to be orthogonal to all initial states leading to misleading states at any time. They form a “very small” Hilbert subspace $\tilde{H}\ll H$, which is extremely restricted compared to H because there are potentially infinitely more orthogonal states that contain invalid records compared to those that contain valid records.*
*For the pilot-wave theory (PWT) and variations [6,7], the wavefunction also never collapses, but it is completed with “hidden variables”, e.g., point-particles with definite positions that resolve the superposition. The initial conditions of the wavefunction are constrained exactly as in the MWI case because the same branching structure is needed to make sure that the configurations of the point particles are stable.*

*In standard quantum mechanics (SQM), the superposition is resolved by invoking the Projection Postulate. This happens when the observation causes a macroscopic effect, usually by changing the pointer of the measuring device. Let ${\left({\hat{P}}_{\alpha }\right)}_{\alpha }$ be a set of mutually orthogonal projectors that correspond to the macro-states, so that $\sum_{\alpha }{\hat{P}}_{\alpha }$ is the identity operator ${\hat{I}}_{H}$ on H. In our case, these projectors are determined by the eigenstates of the pointer observables, so they are ${\hat{I}}_{H_{S}}⊗|\zeta_{j}^{A}\rangle \langle \zeta_{j}^{A}\left|⊗\right|\zeta_{k}^{B}\rangle \langle \zeta_{k}^{B}|⊗{\hat{I}}_{H_{E}}$, or some other projectors finer than them, obtained by including other observables that commute with the pointer observables.*

*The Projection Postulate makes it possible that, for any state at $t_{2}$, there are many possible initial states that can lead to it by unitary evolution alternated with projections. Let us verify that any misleading state at $t_{2}$ can be obtained like this. Suppose that the state vector is projected at a time t from $\Psi_{1}$ to $\Psi_{2}$. Since $|\langle \Psi_{2}|\Psi_{1}\rangle {|}^{2}={\left|\langle \Psi_{1}|\Psi_{2}\rangle \right|}^{2}$, the probability that $\Psi_{1}$ projects to $\Psi_{2}$ in forward time evolution equals the probability that $\Psi_{2}$ projects to $\Psi_{1}$ in backward time evolution. This implies that if we propagate a misleading state at $t_{2}$ back in time to $t_{i}$ “unprojecting” whenever is needed, we should find a set of initial states at $t_{i}$ that can evolve forward in time (with projections) into the misleading state at $t_{2}$ (where I define the result of the “unprojection” of a vector |Ψ〉 with the projector operator ${\hat{P}}_{\alpha }$ as the set of vectors $|\Psi^{\prime }\rangle$ so that $\langle \Psi |{\hat{P}}_{\alpha }|\Psi^{\prime }\rangle \neq 0$.) All these initial states should, therefore, be extremely unlikely. Since any initial state that is not orthogonal to all of them has components that can evolve into misleading states, these should be extremely unlikely as well. This, again, constrains the possible states to an extremely restricted subspace $\tilde{H}$ orthogonal to all initial states that could evolve into misleading states at any future time.*

*In collapse theories [8,9], spontaneous localization is not defined in terms of projectors but by multiplying the wavefunction with a Gaussian function centered at a random point in the configuration space. This happens at random times. Gaussian functions do not form an orthonormal basis, but they partition the identity, so the possible histories with collapses at the same moments of time also add up to the identity, and we can apply similar reasoning as in the SQM case, obtaining the same conclusion. Moreover, when the wavefunction is multiplied by a Gaussian, the result has tails, and decoherence is required to prevent those tails from interfering because otherwise, the collapse could make the state jump to a too different state, so similar constraints as in MWI are required.*

*Therefore, in all cases, the Born rule constrains the states to an extremely restricted subspace $\tilde{H}$ of H, and Belief 1 (Universality) is contradicted.*
**Remark** **1.**
*In all cases, the non-misleading states are constrained to an extremely restricted subspace $\tilde{H}\ll H$. In fact, to protect the Born rule at arbitrary times in the future, the constraints must be valid at all times. But the subspace ${\tilde{H}}_{0}\ll \tilde{H}$ of non-misleading initial states that can lead to states in $\tilde{H}$ is even more restricted than $\tilde{H}$.*
**Refutation** **2**(of Belief 2). *Consider a factorization $H=H_{1}⊗H_{2}$, obtained by dividing the total system into a subsystem $S_{1}$ and the rest of the world, $S_{2}$. The tensor-product basis cannot have all its elements in $\tilde{H}$, because then H would be included in $\tilde{H}$. Therefore, there are tensor-product states that are forbidden or at least extremely unlikely, contrary to Belief 2. Interestingly, even if $S_{1}$ consists of a single particle, the Subsystem Independence is violated. This also contradicts Statistical Independence from Definition 1.***Refutation** **3**(of Belief 3). *From Observation 1, there are states containing invalid records. As seen, avoiding them requires fine-tuning that violates Beliefs 1 and 2.***Challenge** **1**(of Beliefs 4 and 5). *To show the independence of ${\tilde{H}}_{0}$ from the dynamical law and the validity of future records, we need to show that modifications ${\hat{H}}^{\prime }$ of the Hamiltonian $\hat{H}$ lead to valid records. The valid records should be consistent with the modified Hamiltonian ${\hat{H}}^{\prime }$. It is not needed to prove this for all possible modifications of $\hat{H}$, but at least for those that are local and preserve its tensor-product decomposition into Hilbert spaces for elementary particles. Partial progress, but not a definitive proof, is described in Section 3.*This concludes the proof of Theorem 1. □

**Corollary** **1.**
*The state of any subsystem S is not completely independent of the state of the rest of the universe, in the sense that there are extremely unlikely and hence practically forbidden states of the form ψ⊗ε, where ψ is the state of S and ε is the state of the rest of the universe.*


**Proof.** See Refutation 2 (of Belief 2). □

Corollary 1 suggests that the tensor-product Hilbert space may contain too many states, and we should consider a much smaller subspace instead.

**Question** **1.**
*Why, in all known examples, does textbook quantum mechanics work without fine-tuning?*


**Answer** **1.**Textbook scenarios of quantum experiments make assumptions that implicitly fine-tune the system:(a) measuring devices already exist, despite their construction being complicated, necessitating precision technology, and depending on the Hamiltonian to function as desired,(b) before measurements, the measuring devices are in the “ready” states, like $\zeta_{0}^{A}$ and $\zeta_{0}^{B}$ in Equations (Equation 2) and (Equation 3),(c) the observed systems and measuring devices are initially in separable states like $\psi ⊗\zeta_{0}^{A}$ in Equation (Equation 2),(d) after the measurement, the pointer states both before and after the measurement are known, so that the recovery of the state of the observed system is possible.All examples, in all interpretations of quantum mechanics, assume (a), (b), (c), and (d), but this requires fine-tuning. Even MWI requires that branching be time-asymmetric, which constrains the initial conditions to ensure (a). Assumption (a) requires the initial conditions to depend on the Hamiltonian. Linear combinations satisfying (b) and (c) are valid, but they form a strict subspace of the full tensor-product Hilbert space H.If, as in the example from the proof of Theorem 1, $\hat{A}=\hat{B}$ and j≠k, the state $\Psi_{2}\left(t_{2}\right)=\psi_{k}^{B}⊗\zeta_{j}^{A}⊗\zeta_{k}^{B}⊗\ldots$ cannot be reached from the state $\Psi_{1}\left(t_{1}\right)=\psi_{j}^{A}⊗\zeta_{j}^{A}⊗\zeta_{0}^{B}⊗\ldots$. This means that the pointer state $\zeta_{j}^{A}$ contained in $\Psi_{2}\left(t_{2}\right)$ is an invalid record since in the histories leading to $\Psi_{2}\left(t_{2}\right)$ there is no measurement of $\hat{A}$ in which the observed system was found at $t_{1}$ in the state $\psi_{j}^{A}$ and the pointer was in the corresponding state $\zeta_{j}^{A}$. Therefore, the misleading state $\Psi_{2}\left(t_{2}\right)$ is excluded simply because the state at $t_{1}$ was assumed to be $\Psi_{1}\left(t_{1}\right)$.Even if we are not aware of this, we usually take for granted records at different times. But all that is available to us are the present-time records of past events. From these records or memories, we infer laws and make predictions about future times, and experiments confirm them. This is possible because the invalid records are already excluded by the constraints of the initial conditions implicit in the assumptions (a), (b), (c), and (d).All these and more become apparent when we try to build a model, particularly (d) makes the necessity of fine-tuning clear, as it will be seen in more detail in Section 3, Problem 2.

Theorem 1 and Corollary 1 pose a puzzle for some common-sense beliefs about quantum mechanics and about the possibility of having reliable records of past events. In Section 3, I will try to make progress toward the resolution of this puzzle and address Challenge 1.

## 3. A Counterexample?

Let us try to address Challenge 1 by building a model.

I start with an ideal model, which seems to need the least fine-tuning possible. Since even general ideal quantum measurements turn out to require very special initial conditions, irreversibility is needed. Therefore, we must refine the model by including the Past Hypothesis and the possibility of bound states. The resulting model suggests that Belief 5 can still be true, but Belief 4 is partially contradicted by the necessity to appeal to the dynamical law.

The first model is designed so that it contains the simplest possible measuring devices, each consisting of a single particle. The pointer state will be represented as an internal state. A measurement requires a limited duration of interaction between the observed system and the measuring device, and this is, in general, achieved by interactions with a limited range. Therefore, the measuring device and the observed system should be able to be well-localized and separated in space before and after the measurement. Then, the model needs a space for positions, internal degrees of freedom, and local interactions.

The first model I propose consists of particles moving in the lattice $Z^{3}$ and interacting when they occupy the same position in the lattice. The time evolution takes place in discrete steps, so t∈Z. A continuous version of this model is possible, but for simplicity, I start with discrete space and time.

Each particle, labeled with j∈J, has attached an internal Hilbert space $V_{j}⊗S_{j}$. It includes a velocity space $V_{j}:=\mathrm{span}{\left(V\right)}^{3}$, where $V={\{|-1\rangle }^{v}{,|0\rangle }^{v}{,|1\rangle }^{v}\}$, and an internal space $S_{j}:=\mathrm{span}{\{|a\rangle }^{m}\left|a\in Z_{m}\right\}$, where $Z_{m}:=\left\{0,1,\ldots ,m-1\right\}$, representing either the pointer states or the internal degrees of freedom to be measured.

Since time is discrete, for the dynamical law, we only need a unitary operator $\hat{U}$ representing the evolution of the system for a unit of time. The free evolution of each particle, labeled by *j*, is given by its own unitary operator
(5)$${\hat{U}}_{j}{|x,v\rangle }_{j}{|a\rangle }_{j}={|x+v,v\rangle }_{j}{\hat{U}}_{j}^{m}{|a\rangle }_{j},$$
where v∈V and $a\in Z_{m}$, and ${\hat{U}}_{j}^{m}$ acts on $S_{j}$.

Please note that in this ideal quantum system, there is no uncertainty trade-off between the positions and momenta, and therefore particles can be well-localized “wave packets”.

When two particles occupy the same position, their dynamics include interaction, which I specify to be
(6)$$\begin{array}{cc} {\hat{U}}_{jk} & |x_{j},v_{j}\rangle_{j}|a_{j}\rangle_{j}^{k}|x_{k},v_{k}\rangle_{k}{|b_{k}\rangle }_{k}^{j}\\ = & |x_{j}+v_{j},v_{j}\rangle_{j}{\hat{U}}_{j}^{m}|a_{j}⊕\delta \left(x_{j}-x_{k}\right)b_{k}\rangle_{j}^{k}|x_{k}+v_{k},v_{k}\rangle {\hat{U}}_{k}^{m}{|b_{k}⊕\delta \left(x_{j}-x_{k}\right)a_{j}\rangle }_{k}^{j}, \end{array}$$
where ⊕ denotes the addition modulo *m*.

This requires some explanations. The Kronecker symbol $\delta (x_{j}-x_{k})=1$ if $x_{j}=x_{k}$ and 0 otherwise (we are in a discrete model) forces the interaction to take place only when the two particles occupy the same position. The upper index *k* for the basis of $S_{j}$ indicates that the basis also depends on the particle *k* with which *j* interacts. The reason for using different bases for the same particle, a basis for each other particle with which it interacts, allows the existence of different incompatible measurements of the same particle *j*. This was already ensured by the existence of ${\hat{U}}_{j}^{m}$, but we want the most general settings. If $x_{j}\neq x_{k}$, the two particles evolve freely, as in Equation (Equation 5).

Interactions should also be defined for the case when more than two particles occupy the same position, but I will discuss only interactions between two particles.

Some of the particles are of a special type, $J_{M}\subset J$, representing measuring devices, while for each $j\in J_{M}$, all the other particles from J (not only those from $J\setminus J_{M}$) are the observed systems. The pointer states are the elements of the basis of $S_{k}$, for each $k\in J_{M}$. The macro-states are characterized by the positions, velocities, and the pointer states of the particles from $J_{M}$, i.e., a triple $\left(x_{k},v_{k},b_{k}\right)$. The basis ${\left(|b_{k}\rangle_{k}^{j}\right)}_{k}$ from Equation (Equation 6) is assumed, only for type $J_{M}$ particles, to be independent of the particles *j* with which they interact. The free evolution operator ${\hat{U}}_{k}^{m}$ for *k* is taken to be the identity operator so that the pointer states are stable in time.

Let us see how an interaction between a particle *k* from $J_{M}$ and another particle *j* constitutes a measurement of *j* by *k*, when $x_{j}=x_{k}=x$ and $v_{j}\neq v_{k}$. If $b_{k}=0$, the pointer state $|b_{k}\rangle_{k}^{j}={|0\rangle }_{k}\in S_{k}$, so the measuring device *k* is in the “ready” state. If the observed particle *j* is in a basis state $|a_{j}\rangle_{j}^{k}$, Equation (Equation 6) gives
(7)$${\hat{U}}_{jk}|x,v_{j}\rangle_{j}|a_{j}\rangle_{j}^{k}|x,v_{k}\rangle_{k}{|0\rangle }_{k}=|x+v_{j},v_{j}\rangle_{j}{\hat{U}}_{j}^{m}|a_{j}\rangle_{j}^{k}|x+v_{k},v_{k}\rangle |a_{j}\rangle_{k}.$$ We see that the particle *j* was measured in the basis of $S_{j}$ corresponding to the measuring device *k*, and the result is recorded as $|a_{j}\rangle_{k}\in S_{k}$. After that, ${\hat{U}}_{j}^{m}$ may change the state of *j*, but it was already measured. The measured observable has as eigenbasis the basis ${\left(|a_{j}\rangle_{j}^{k}\right)}_{a_{j}\in Z_{m}}$.

Now, let us see how it works. We assume that at the initial time $t_{i}=0$, the particles have well-defined positions. Equation (Equation 6) ensures us that they remain well-defined. The pointers decompose the total state so that each pointer is in a definite pointer eigenstate. In SQM, we invoke the Projection Postulate, while in MWI, the total state branches. When two particles occupy the same point in space, they interact. If one of them is a measuring device, this leads again to the superposition of macro-states, demanding again the invocation of the preferred solution to the measurement problem.

At this point, this model seems to contain all that is needed to describe a quantum world, and we may be tempted to proclaim that the problem was solved:

**Solution** **1**(Premature claim). *In this simple ideal model, it seems that the fine-tuning is minimal, and the conspiracy is avoided. The only fine-tuning needed is that the initial state consists of particles with definite positions. Then, each interaction with a measuring device (particle from $J_{M}$) constitutes a measurement. The positions of the measuring devices, along with their pointer states, can be used to characterize the macro-states. Moreover, if we change the free evolution operators ${\hat{U}}_{j}^{m}$ and the bases from the interaction evolution operators in Equation (Equation 6), the system evolves properly, leading to different histories, whose states contain different records, but these records are consistent with the history of the system. Therefore, the initial state does not have to depend on the dynamical law or of the future records since we can change it and still have only compatible records.*

However, two problems show that the claims from Solution 1 would be rushed. These problems are not specific to this model. They must be solved for any other model.

The first one is particular to interpretations requiring decoherence, like MWI and PWT:

**Problem** **1.**
*There is no definite branching structure.*


The terms in superposition, corresponding to different macro-states, can evolve so that they contain the same pointer states, which means that the branches interfere or even are joined back into a single branch. For example, consider the interaction
(8)$$\begin{array}{cc} {\hat{U}}_{jk}|x,v_{j}\rangle_{j}|a_{j}{⊖1\rangle }_{j}^{k}|x,v_{k}\rangle_{k}{|1\rangle }_{k}\; & =|x+v_{j},v_{j}\rangle_{j}{\hat{U}}_{j}^{m}|\left(a_{j}⊖1\right){⊕1\rangle }_{j}^{k}|x+v_{k},v_{k}\rangle |\left(a_{j}⊖1\right){⊕1\rangle }_{k}\\ \; & =|x+v_{j},v_{j}\rangle_{j}{\hat{U}}_{j}^{m}|a_{j}\rangle_{j}^{k}|x+v_{k},v_{k}\rangle |a_{j}\rangle_{k}, \end{array}$$
where ⊖ denotes the subtraction modulo *m*.

Here, the final macro-state, the state of the measuring device, is $\left(x+v_{k},v_{k},a_{j}\right)$, and it is the same as that from Equation (Equation 7), but the macro-state immediately before the interaction was $\left(x,v_{k},1\right)$, so it was different from the prior macro-state from Equation (Equation 7), which was $\left(x,v_{k},0\right)$. This means that two macro-states can interfere. A superposition of the two prior macro-states from Equations (Equation 7) and (Equation 8) will evolve into the same macro-state. In other words, branching happens towards the past in this example. The branching structure is not a tree with the trunk in the past and the branches in the future, as it should be. Since the interference of separate branches spoils the Born rule, we need to fix this by choosing suitable initial conditions.

Another problem, more important, is that

**Problem** **2.***At any given time, the pointer states contain insufficient information to encode the states of the observed systems. This is true even for more sophisticated models of quantum measurements, as long as the state of the pointer can be known only by observing it*.

Let us see why. If the pointer state is not in the “ready” state ${|0\rangle }_{k}$ before the measurement, the particle *k* performs a disturbing measurement, and the pointer no longer indicates the state of the observed particle,
(9)$${\hat{U}}_{jk}|x,v_{j}\rangle_{j}|a_{j}\rangle_{j}^{k}|x,v_{k}\rangle_{k}|b_{k}\rangle_{k}=|x+v_{j},v_{j}\rangle_{j}{\hat{U}}_{j}^{m}|a_{j}⊕b_{k}\rangle_{j}^{k}|x+v_{k},v_{k}\rangle |a_{j}⊕b_{k}\rangle_{k}.$$

Therefore, while the pointer correctly indicates the state of the observed system after the measurement, the interaction disturbed it, and the observed system’s state before the measurement is now unknown, even if it was an eigenstate. To know the value $a_{j}$ before the measurement, we need to know the pointer’s state $|b_{k}\rangle_{k}$ before the measurement. Our intuition may fool us into thinking that both $b_{k}$ and $a_{j}⊕b_{k}$ can be known, but at the time when $a_{j}⊕b_{k}$ is known, $b_{k}$ is no longer available. To make it available, the pointer of *k* must be measured by another measuring device $k^{\prime }\in J_{M}$. But the measuring device $k^{\prime }$ has the same problem since we would have to know its pointer state before and after it interacts with the measuring device *k*. This leads to an infinite regress. This happens for all models of measurements, as long as it is assumed that the pointer state can be known only by direct observation.

Albert wrote about such situations in [10], p. 118:


*There must […] be something we can be in a position to assume about some other time–something of which we have no record; something which cannot be inferred from the present by means of prediction/retrodiction–the mother (as it were) of all ready conditions. And this mother must be prior in time to everything of which we can potentially ever have a record, which is to say that it can be nothing other than the initial macrocondition of the universe as a whole.*


He is talking about what he calls the *Past Hypothesis*, the condition proposed by Boltzmann [11,12] to explain the Second Law of Thermodynamics, that the universe was a very long time ago in a very low-entropy state.

But how can this allow the measuring device $k^{\prime }$ to measure the previous state of the measuring device *k*? For example, in practice, a photographic plate can be used to record the position of a particle. We do not need to observe the photographic plate before the experiment. We prepare it, assuming that thermalization does the job. The same is true for a bubble chamber. Eventually, any real-life measurement requires the Past Hypothesis.

So, let us assume that some of our particles act like a gas or a thermal bath that brings the pointer state of the measuring devices to the ready state, assumed to be its energy ground state. The time interval needed to reach equilibrium should allow the record to be preserved long enough, and the pointer state is more stable in its ground state. With properly chosen parameters, this may solve our Problem 2. Let these particles form a subset $J_{E}\subset J\setminus J_{M}$. We assume $J_{E}$ to make the overwhelming majority of the particles from J so that each measuring device is involved, between two measurements, in many weak interactions with the particles from $J_{E}$.

At this point, our model turns out, again, to be too simple. Different couplings require different interaction times, so the interaction times should vary more. The velocities should also vary to account for the entropy. However, the model can be made more realistic by modifying it so that each interaction can have a different range, requiring a longer time to change the pointer state into another eigenstate by thermalization.

Another problem is that the interaction between the particles from $J_{E}$ and those from $J_{M}$ are, like any interaction (Equation 6), time-reversible. We recall that the decay of excited atoms or particles with finite lifetime is perhaps the simplest irreversible process. In the Weisskopf-Wigner approximation [13], an excited atom evolves into a superposition of an excited atom state and a ground atom-plus-photon state, and the coefficient of the excited term decays exponentially. The irreversibility is due to the fact that the emitted photon quickly moves away from the atom. A time-reversed situation is possible, of course, and consists of the photon being absorbed by the atom, which becomes excited. But the Past Hypothesis makes the events of absorption of a photon coming from a distance become, in time, less frequent than the emission events, so the most stable state becomes the ground state.

**Solution** **2**(More realistic). *Our model can be made more realistic by including, along with weak couplings with the thermal bath, bound states that constitute the pointers of the measuring device so that the bath changes each pointer state to the “ready” state after a reasonable time. This should allow the Past Hypothesis to be applicable to the model, which should allow the records to be kept like in the real world, likely solving Problem 2. A thermal bath may also solve Problem 1, by environmental decoherence [14,15,16].*

**Observation** **4**(on Solution 2). *We have seen that the original ideal model is unable to extract the states of the observed systems and record them. A more realistic version requires the Past Hypothesis and measuring devices constructed of bound states. However, the updated model needs weak couplings and bound states with carefully chosen parameters, so it depends on the dynamical law. By changing the Hamiltonian, the measuring devices can stop working as such, and the observables defining the macro-states may no longer do this job. The Past Hypothesis should not merely state that the initial macro-state or the initial subspace ${\tilde{H}}_{0}$ was extremely small, but also that the dynamical law matters in the choice of this region. This implies that Belief 4 is contradicted at least partially, and Challenge 1 is still open.*

## 4. Past Hypothesis

**Question** **2.**
*It is thought that the Past Hypothesis and the Statistical Postulate are sufficient to ensure the reliability of the records. Does Theorem 1 challenge this?*


**Answer** **2.**There are numerous good arguments that the Past Hypothesis (PH) and the Statistical Postulate together are sufficient to explain the reliability of the records. An incomplete selection that cannot do justice to the literature is [10,17,18,19,20,21,22]. Critiques of the Boltzmann program can be found e.g., in [23,24].Usually, the Past Hypothesis is formulated as the request that the initial state should be confined to a very restricted initial Hilbert subspace or a relatively very small region of the phase space. This, according to Boltzmann, would ensure the very low entropy of the initial state, necessary for the statistical account of the Second Law of Thermodynamics. But Theorem 1 and Corollary 1 imply that there is more to the story, in particular, that the Past Hypothesis must depend on the dynamical law and violate Statistical Independence or be extended with an additional condition or a law that guarantees these conditions. The attempt from Section 3 to build a model that escapes Theorem 1 confirms this and may suggest how the dynamical law should intervene in the Past Hypothesis, even in a model that is intentionally built to separate as much as possible the initial conditions from the dynamics.

**Question** **3.**
*In an infinitely large universe or a multiverse, with an eternal history, this apparent fine-tuning happens with necessity somewhere or sometimes, albeit extremely rarely. Since intelligent living beings like us need reliable memories to exist and survive, they can only exist in such a region. Is this not enough to explain away the fine-tuning?*


**Answer** **3.**If we were in such a region where it just happened that the Metaprinciple NMU was respected up to a time $t_{1}$, it would be extremely more likely that NMU will be violated than not after $t_{1}$. Subsystems able to record events, like measuring devices in the “ready” state, are a limited resource. But then, we should already observe that the Born rule “wears off” and the world becomes flooded by unreliable records, becoming increasingly inconsistent, like a dream. Also, by an indexical argument, if we rely on anthropic reasoning, it would be much more likely that an observer finds herself as a Boltzmann brain in a region where NMU is violated.

**Question** **4.**
*Is it not the case that these results apply to classical physics as well?*


**Answer** **4.**This is correct. This article is about quantum mechanics because the world is quantum, but the main idea applies to classical physics, too. The following result proves it.

**Corollary** **2.**
*The conclusion of Theorem 1 applies to classical physics, too.*


**Proof.** As shown by Koopman and von Neumann, classical physics can be formulated as a quantum theory [25,26,27]. In Koopman’s quantum representation of a classical theory, the wavefunctions are defined on the classical phase space (not on the classical configuration space as we do when quantizing a classical system). Each classical state (q,p), where $q=(q^{1},q^{2},\ldots )$ are the generalized positions and $p=(p_{1},p_{2},\ldots )$ the generalized momenta, is then represented by a Dirac delta distribution, represented as a state vector by |q,p〉, and the dynamics preserves classicality, i.e., it evolves states of the form |q,p〉 into states of the same form. The classical states |q,p〉 have both definite positions and definite momenta, and this is preserved in the quantum representation, where $\hat{q}$ and $\hat{p}$ commute (because ${\hat{p}}_{j}$ and $-i\hbar \frac{\partial \;}{\partial q^{j}}$ are different operators). Since all classical observables are functions f(q,p) of the positions and momenta, the operators representing them, $f(\hat{q},\hat{p})$, also commute, and |q,p〉 are eigenstates for $f(\hat{q},\hat{p})$ too. So classical physics can be formulated as a quantum theory in which all relevant observables commute (note that the operator $-i\hbar \frac{\partial \;}{\partial q^{j}}$ does not correspond to any classical observable, it is not ${\hat{p}}_{j}$, so it is not part of the theory, and there is no use for the Planck constant). The only vectors of interest in the Hilbert space H are those representing classical states of the form |q,p〉. Superpositions of the basis vectors can be used not to represent states but classical ensembles.The proof of Theorem 1 is limited to commuting observables and shows that the set of allowed states must be the classical vectors |q,p〉 from a very restricted Hilbert subspace ${\tilde{H}}_{0}\ll \tilde{H}$. Returning to the original classical theory, this corresponds to a very small, zero-measure subset of the classical phase space. □

Therefore, except for the parts involving non-commuting observables, the entire discussion from this article applies to classical physics as well.

## 5. Statistical Independence

**Question** **5.**
*Does Corollary 1 refute the Statistical Independence assumption used by Bell in his proof that any interpretation of quantum mechanics in which measurements have definite outcomes must be nonlocal [2,28]?*


**Answer** **5.**Short answer: no, but before addressing this question, let us establish that the experiment of interest in Bell’s theorem is covered in the proof of Theorem 1.

**Example** **1**(EPR). *The Einstein–Podolsky–Rosen (EPR) experiment [29] is a particular case of the example from the proof. At $t_{1}$, the preparation results in a singlet state of two entangled spin 1/2 particles, with total spin 0. The preparation is made by taking as $\hat{A}$ an observable that has the singlet state among its eigenstates and pre-selecting this state. At $t_{2}$, Alice measures the spin of the first particle along a direction a, and Bob, in a different place, measures the spin of the second particle along a direction b. Let ${\hat{S}}_{a}$ and ${\hat{S}}_{b}$ be the two spin observables. Since the two measurements are performed on different particles, they commute, $\left[{\hat{S}}_{a}⊗{\hat{I}}_{2},{\hat{I}}_{2}⊗{\hat{S}}_{b}\right]=0$, and can be seen as a single measurement of the observable $\hat{B}={\hat{S}}_{a}⊗{\hat{S}}_{b}$ performed on the pair of particles. If a=b, the misleading outcomes are those resulting in parallel spins. But the states containing pointer states that correspond to these results are valid vectors in H. So, the initial conditions that can lead to these states have to be forbidden, which means that SI from Definition 1 must be violated.*

If, in Definition 1, Pr{B}>0, SI is equivalent to Pr{A|B}=Pr{A}, the form that appears in Bell’s theorem [2]. But Bell’s SI refers to the independence of the observed system from the measurement settings, and it is not refuted by Corollary 1. Corollary 1 allows the observed system to be in any state $|\psi \rangle \in H_{1}$, provided that the rest of the world |ε〉 is restricted to a strict subset of its Hilbert space $H_{2}$. In Bell’s SI, the only considered states $|\varepsilon \rangle \in H_{2}$ are those containing Alice and Bob’s measuring devices. Moreover, the settings of the measuring devices have macroscopic properties that can be realized in infinitely many ways at the microscopic level. This means that |ε〉 can be chosen in many ways that are compatible with the macroscopic settings of the measuring devices. To violate Bell’s SI assumption, all possible microscopic ways to realize the same measurement settings should be excluded, and Corollary 1 does not do this.

This is why Corollary 1 does not contradict Bell’s SI, while still contradicting the SI from Definition 1. The EPR experiment is a particular case of the situation from the proof of Theorem 1 (Example 1).

**Question** **6.**
*Does Theorem 1 imply superdeterminism?*


**Answer** **6.**Theorem 1 does not refute nonlocal interpretations like PWT or GRW, or multiple-outcome interpretations like MWI. However, if SI is a reason to reject superdeterministic approaches, Corollary 1 shows that all interpretations are guilty of the same sin, but to a lesser extent. There is still a large difference of degree between violating the SI from Definition 1 and violating Bell’s SI.

**Question** **7.**
*Isn’t the violation of Statistical Independence unscientific [21,30,31,32]? By fine-tuning you can make the theory predict anything. According to Maudlin:*

*If we fail to make this sort of statistical independence assumption, empirical science can no longer be done at all.*
Bell wrote ([33], p. 244):
*In such ‘superdeterministic’ theories the apparent free will of experimenters, and any other apparent randomness, would be illusory. Perhaps such a theory could be both locally causal and in agreement with quantum mechanical predictions. However I do not expect to see a serious theory of this kind.*



**Answer** **7.**Now we have a theorem showing that quantum mechanics itself requires the violation of SI (Definition 1) but not of Bell’s SI (Answer 5).We cannot do science without the possibility to trust the records of past experiments and our own memory. And this, as we have seen, requires SI violations. Does this mean that we can no longer do science?There are proposals that try to save locality by sacrificing Bell’s SI [34,35,36,37,38]. Other proposals even try to save locality but also to maintain unitary evolution with a single world, i.e., without branching, and without collapse or “hidden variables” [39,40,41]. These approaches require very special initial conditions [42]. But, as Theorem 1 shows, so does quantum mechanics in all interpretations, and it also violates SI from Definition 1.However, it is unfair to say that you can predict anything by fine-tuning, because approaches with unitary evolution and a single world could so far only reproduce very simple quantum experiments. In addition, as explained in [39,41], these approaches make very strict predictions, for example, that the conservation laws hold even if their violation is allowed by SQM [41,43] and other interpretations, including in MWI (per branch).

## 6. New Law?

**Question** **8.**
*Do you have another explanation?*


**Answer** **8.**Maybe there is a still unknown law that restricts the possible states to $\tilde{H}$. Since subsystems are not independent (Corollary 1), the tensor-product Hilbert space is too large and should be replaced by its subspace $\tilde{H}$. Since the restrictions do not depend on time (Remark 1), $\tilde{H}$ should be an invariant subspace under unitary evolution. This justifies the hypothesis that there is an as-yet-unknown law that specifies what kinds of states are allowed and extends the Past Hypothesis to include the dependence of the allowed Hilbert subspace $\tilde{H}\ll H$ on the dynamical law. This may be a superselection rule, similar to the superselection rules that forbid superpositions of systems with different electric charges or different spins [44]. If such a law or superselection rule exists for $\tilde{H}$, it could explain the SI violations without fine-tuning. But such a law would have to encode the dependence of $\tilde{H}$ of the Hamiltonian and the macro projectors ${\left({\hat{P}}_{\alpha }\right)}_{\alpha }$.

Therefore, the challenge is

**Challenge** **2.**
*Find a simple universal law that describes the restriction of the Hilbert space to the subspace $\tilde{H}$ from the proof of Theorem 1 without fine-tuning. Then verify whether this law violates Bell’s SI or only the general SI from Definition 1.*


It remains to be seen whether or not this new law restricts the Hilbert space more than is needed for standard quantum mechanics so that Bell’s SI is violated sufficiently to allow unitary evolution to resolve the measurements without collapse, as suggested in proposals like [39,40,41].

## Data Availability

No new data were created or analyzed in this study. Data sharing is not applicable to this article.

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
