# Peer review of "Does Quantum Mechanics Require “Conspiracy”?"

_entropy, 2024, doi:10.3390/e26050411_

Round 1

Reviewer 1 Report

Comments and Suggestions for Authors

It is well known from general, statistical-mechanical considerations that one must assume some kind of special boundary conditions ("Past Hypothesis") for the initial state of the universe to maintain the reliability of records (Albert, 2000). It is interesting to explore this issue and its ramifications in more detail in the context of quantum mechanics, and the paper makes some interesting points to that effect. But many of the arguments remain too unclear or uncompelling. 

1) The paper argues that, according to standard QM, initial states can evolve (with projections) into a "forbidden" state at t_2. Indeed, they can, but not by projections that correspond to repeated, projective measurements. For other types of measurements, the respective final state is not necessarily "forbidden" though. The argument thus remains unclear.

2) It is not correct that Bohmian mechanics adds another layer of fine-tuning to justify the Born rule. On the contrary, the Born rule holds for typical initial particle configurations (see Dürr, D., Goldstein, S. & Zanghí, N. Quantum equilibrium and the origin of absolute uncertainty. J Stat Phys 67, 843–907 (1992)). 

3) To justify the "Metaprinciple NMU (Non-Misleading Universe)", would it not suffice that an evolution into "forbidden" states is very unlikely rather than impossible? In any physical theory, atypical 'fluctations' could lead to misleading records. 

4) Much of the discussion conflates statistical independence with what the the paper calls "Subsystem Independence". In quantum mechanics, statistical independence only requires that two systems are in a product state, not that they could be in any possible combinations of states. One might also discuss statistical independence on the ensemble level (represented by mixed states), but this seems like a different issue. 

5) In general, the fact that two systems have some common constraints in the past does not mean that they cannot be statistically independent -- at least "independent enough" for all practical purposes. In particular, chaotic dynamics tend to produce (some kind of) statistical independence in complex systems. This is a very subtle issue, but the kind of issue one regularly deals with in statistical mechanics. 

6) The paper asks whether it  Cor. 1 refutes the statistical independence assumption in Bell's theorem. The answer is simply NO. No combination of singlet state and parameter settings is forbidden, and Cor. 1 says nothing about additional variables over and above the quantum state. 

7) "Branching is reversible." Microscopically, yes. In a thermodynamic sense, no. This would require a more nuanced discussion. 

I think the paper overall needs to be refocused. While it makes some good and thought-provoking points, it tries to make too many different points at once. 

Author Response

Dear Reviewer 1,

Thank you very much for the thought-provoking comments. Please find attached a pdf with the author's answers to the comments and suggestions made by the Reviewers, and the manuscript text with the changes integrated and highlighted.

Best regards,

Author

Reviewer 2 Report

Comments and Suggestions for Authors

This well-written paper strikes me as thought provoking. Its core argument is that the standard account of ideal measurement in quantum mechanics predicts certain correlations between the results of consecutive measurements, whereas states violating these correlations exist as well in Hilbert space. Projecting the latter type of states back in time, we arrive at initial states whose time evolution exhibits incorrect correlations and thus violations of the Born rule. Therefore, some selection principle for initial states must be assumed to make sense of our actual quantum practice.

The argument as given is interesting, in my opinion. But I wondered whether the conclusions were fully supported  by it, and I also have some questions about further details.

Starting with the latter:  What makes this argument typical quantum mechanical? There is a lot of emphasis on Hilbert space, unitary evolution, different interpretations of QM, but at first sight the argument can be given in nearly identical form in the context of classical mechanics. Repeated ideal measurements should give identical results, but points in phase space where this is not the case surely exist. Evolving these points back in time leads to initial states that lead to "unreliable measurement results".  In many cases this will be processes that would normally not be considered measurements at all, but rather the fortuitous grouping together of atoms and molecules in the form of a device and records. So, what is the new element introduced in this type of argument by QM? At first sight the situation in QM is exactly the same. 

Related to this: in my opinion it is questionable whether it is justified to call such strange initial states "forbidden". Isn't the standard attitude that processes of the strange kind can occur, and actually will occur in the universe, but are very rare in our environment?

A related remark with respect to "belief 1". I think that the wording of this principle is too strong. Normally one would only have a belief of this kind for measurements that start in ready states etc, with exclusion of Boltzmann brain situations. It remains true, though, that there is the epistemic problem of how to know, reliably, that such a real measurement situation with ready states to start with is actually present.

Concerning the question of what to conclude from the argument: It is of course true that following non-Born states back in time, we arrive at initial states that show exactly this non-Born behavior at the instant we started with. But this in itself does not show that the total history coming from the initial state in question is typically non-Born. Perhaps, given decoherence etc, even these strange initial states give rise to histories of the universe that for considerable parts are "Born-normal". Perhaps the argument can be generalized to cover this point, I am not totally sure. Anyway, it would be nice to see some discussion of this. In my opinion one can only say that the strange initial states are to be forbidden if they lead to evolutions that conflict with the Born rule in a general way, and not merely in the form of infrequent Boltzmann fluctuations.

I think the paper is thought-provoking and of interest, but could use some further analysis.  

Author Response

Dear Reviewer 2,

Thank you very much for the thought-provoking comments. Please find attached a pdf with the author's answers to the comments and suggestions made by the Reviewers, and the manuscript text with the changes integrated and highlighted.

Best regards,

Author

Round 2

Reviewer 1 Report

Comments and Suggestions for Authors

I appreciate the revision and find the manuscript improved. However, my objections have not been fully resolved. 

1. While I agree that "misleading states" is better terminology than "forbidden states," this wasn't really the point. The point is that a state like (3) with A=B and j \neq k is not misleading per se. It's simply a state with two measurement devices whose pointers are pointing in different directions. It would be misleading as a record of a repeated reproducible measurement. However, according to standard QM, it cannot result from such a measurement (or only with very low probability in the sense explained on p. 4 of the paper). If the state results from a different time-evolution or measurement-like process (including projections), it's not misleading -- or at least not obviously so. For example, (3) can arise from a history in which only the first measurement takes place, and the second device is in the stationary state \xi_k all along. What would be problematic about that? 

2. Regarding the issue of statistical independence, the paper now emphasizes an apparent conflict between the Born rule and a uniform probability distribution over initial states. But this point also remains unclear. 

a) While I get the idea of such an initial probability distribution, it is not usually part of quantum mechanics and insufficiently motivated in the paper. 

b) If I am not mistaken, the probability of a separable (non-entangled) state would be zero according to such a distribution. Hence, this statistical hypothesis doesn't seem to capture the paper's idea of statistical independence to begin with. 

c) In standard QM, the Born rule is best understood as providing transition probabilities rather than probability distributions at a time. With this understanding, it's not clear where the conflict lies. In Bohmian mechanics, the Born rule does provide an actual probability distribution, but, in the first place, for particle configurations, not for quantum states. 

Reviewer 2 Report

Comments and Suggestions for Authors

I still believe the paper is thought-provoking; it is less original than I first thought. I think it is good to have a paper having this after all well-known subject, but now focused on QM, in the literature. 
